# High-Dose Vitamin B6 (Pyridoxine) Displays Strong Anti-Inflammatory Properties in Lipopolysaccharide-Stimulated Monocytes

**DOI:** 10.3390/biomedicines11092578

**Published:** 2023-09-19

**Authors:** Kathleen Mikkelsen, Narges Dargahi, Sarah Fraser, Vasso Apostolopoulos

**Affiliations:** Immunology and Translational Research Group, Institute for Health and Sport, Werribee Campus, Victoria University, Melbourne, VIC 3030, Australia; kathleen.mikkelsen@vu.edu.au (K.M.); narges.dargahi@live.vu.edu.au (N.D.); sarah.fraser@vu.edu.au (S.F.)

**Keywords:** vitamin B6, pyridoxine, broad-spectrum anti-inflammatory, monocytes, inflammatory disease, nutraceutical agent

## Abstract

Vitamin B6 is shown to have anti-inflammatory properties, which makes it an interesting nutraceutical agent. Vitamin B6 deficiency is well established as a contributor to inflammatory-related conditions, whilst B6 supplementation can reverse these inflammatory effects. There is less information available regarding the effects of high-dose vitamin B6 supplementation as a therapeutic agent. This study set out to examine the effects of high-dose vitamin B6 on an LPS-stimulated monocyte/macrophage cell population via an analysis of protein and gene expression using an RT2 profiler PCR array for Human Innate and Adaptive Immune responses. It was identified that high-dose vitamin B6 has a global anti-inflammatory effect on lipopolysaccharide-induced inflammation in monocyte/macrophage cells by downregulating the key broad-spectrum inflammatory mediators *CCL2*, *CCL5*, *CXCL2*, *CXCL8*, *CXCL10*, *CCR4*, *CCR5*, *CXCR3*, *IL-1β*, *IL-5*, *IL-6*, *IL-10*, *IL-18*, *IL-23-a*, *TNF-α*, *CSF2*, *DDX58*, *NLRP3*, *NOD1*, *NOD2*, *TLR-1 -2 -4 -5 -7 -8 -9*, *MYD88*, *C3*, *FOXP3*, *STAT1*, *STAT3*, *STAT6*, *LYZ*, *CASP-1*, *CD4*, *HLA-E*, *MAPK1*, *MAPK8 MPO*, *MX-1*, *NF-κβ*, *NF-κβ1A*, *CD14*, *CD40*, *CD40LG*, *CD86*, *Ly96*, *ICAM1*, *IRF3*, *ITGAM*, and *IFCAM2*. The outcomes of this study show promise regarding vitamin B6 within the context of a potent broad-spectrum anti-inflammatory mediator and could prove useful as an adjunct treatment for inflammatory-related diseases.

## 1. Introduction

Vitamin B6 (pyridoxine) is establishing a reputation in the world of immunology as a molecule of interest due to its potent anti-inflammatory and antioxidant properties. Vitamin B6 exists in several related forms, including pyridoxine, pyridoxal, and pyridoxamine, all of which can be converted into active coenzyme forms in the body [1]. These coenzyme forms are essential for several enzymatic reactions involved in metabolism. The active coenzyme forms of vitamin B6, pyridoxal 5′-phosphate in particular, play an important role in amino acid metabolism. It is also involved in the synthesis of neurotransmitters in the brain (serotonin, dopamine, and gamma-aminobutyric acid), the synthesis of non-protein compounds (sphingolipids and nucleotides), is required for the formation of haemoglobin, and breaks down glycogen [1]. Vitamin B6 is also imperative for the proper functioning of the immune system, aiding in the production of antibodies and cytokines.

It is well established that vitamin B6 deficiency contributes to inflammation and inflammatory disease, whilst vitamin B6 supplementation, in deficiency states, can reverse these effects [2,3,4]. Less is known, however, about the effect of high-dose vitamin B6 as a therapeutic agent and its potential to be developed as a broad-spectrum anti-inflammatory in conditions such as sepsis or cytokine storm, as is seen in COVID-19 patients [5].

Low blood serum B6 is frequently noted in patients with high inflammatory markers [6]. In fact, numerous inflammatory diseases were correlated with vitamin B6 deficiency including atherosclerosis and cardiovascular disease [7,8,9,10,11], rheumatoid arthritis [12,13], inflammatory bowel disease [14,15,16,17,18], type-2 diabetes [19,20,21,22,23,24,25], non-alcoholic fatty liver disease [26,27], and cancer [7,8,21,28,29,30,31]. Vitamin B6 deficiency can disrupt immune response by decreasing the production of protein and nucleic acids, inhibiting immune cell function, and interfering with the metabolic machinery of cells [3]. Furthermore, vitamin B6, as a key player in one-carbon metabolism, is involved in methylation processes, which, when disrupted as occurs frequently in deficiency states, can cause an increase in homocysteine, resulting in vascular and systemic inflammation [32,33]. Inflammation is normally self-limiting with pro-inflammatory reactions followed by anti-inflammatory action, creating a balance within the process of resolving the initial inflammatory trigger, but chronic inflammation is often harder to resolve with the inflammatory microenvironment becoming more complex as the inflammation progresses [34]. Vitamin B6 is different from several other anti-inflammatory compounds in that it functions as a coenzyme and directly targets inflammatory pathways. Standard anti-inflammatory drugs such as non-steroid anti-inflammatories (i.e., ibuprofen) and corticosteroids (prednisone) function by inhibiting specific enzymes or molecules in inflammation, whereas vitamin B6 does not inhibit these molecules but rather supports overall immune function. In fact, vitamin B6 is anti-inflammatory by reducing the accumulation of sphingosine-1-phosphate in a sphingosine-1-phosphate lysate-dependent manner [35], which is not seen by standard anti-inflammatory drugs.

Monocytes and macrophages are key players in the inflammatory process and are the first line of defence against infection caused by bacteria, viruses, and other microorganisms. Monocytes are particularly sensitive to lipopolysaccharides (LPS), which is the major component of Gram-negative bacteria cell walls. LPS is useful for studying the effects of inflammation in monocytes and macrophages due to the profusion of inflammatory mediators triggered via LPS stimulation. This study sought to investigate the effects of a high dose of vitamin B6 on the inflammatory process in LPS-stimulated monocytes, via the analysis of protein and gene expression. It was found that a high dose of vitamin B6 in cell culture exhibited a broad mediation of many inflammatory cytokines, chemokines, pattern recognition receptors, cell surface markers, and other genes involved in inflammatory and defence responses. In particular, inflammatory mediators implicated in hyperinflammatory conditions related to LPS involvement such as ‘cytokine storm’ in patients with sepsis and/or COVID-19 were significantly downregulated in vitamin B6 cultured samples. These include interleukin (*IL)-6, IL-1β, TNF-α, NF-κB*, and Toll-like receptor *(TLR)-4*. The evidence from this study suggests that high-dose vitamin B6 may be useful as a broad-spectrum anti-inflammatory agent and a possible adjunct to current anti-inflammatory treatments.

## 2. Materials and Methods

### 2.1. Cell Culture and Reagents

#### 2.1.1. Culture of U937 Cells

The U937 cell line, which was originally isolated from a 37-year-old male patient with histiocytic lymphoma, is commonly used to study the behaviour and differentiation of monocytes in vitro. They can be differentiated into monocyte/macrophage cells following culture with certain stimulants such as phorbol 12-myristate 13 acetate or vitamin D_3_ [36]. The U937 cell line was purchased from ATCC by the Monash University Department of Immunology. U937 cells are immortal and allow for multiple passages without transformation; however, we made sure to keep the number of passages to a minimum. The U937 cells were cultured in RPMI 1640 media supplemented with 10% heat-inactivated foetal bovine serum (FBS; Sigma-Aldrich, Melbourne, VIC, Australia), 2 mM L-glutamine (Sigma-Aldrich), 100 mg/mL penicillin, and 100 µg/mL streptomycin (Sigma-Aldrich) at 37 °C and 5% CO_2_. The culture media was changed every 3–4 days and cells were passaged accordingly. Once 80–90% confluent, the cells were used in the experiments.

#### 2.1.2. Preparation of Reagents and Treatment of Cells

A stock solution of Pyridoxine Hydrochloride (Vitamin B6-P6280-100G) was freshly prepared according to the manufacturer’s instructions (Sigma, Melbourne, Australia) in phosphate-buffered saline (PBS). Vitamin B6 stock was filtered using a 0.2-micron filter and used in the culture at a final concentration of 250 μg/mL as determined by the IC50 of dose–response experiments of vitamin B6 in MTT assays not shown. Lipopolysaccharide was dissolved in PBS and used at a final concentration of 1 μg/mL overnight to induce inflammation of the monocyte/macrophage cells; this dose was used because it shows inflammation of cells, and several publications have used this dose to induce inflammation to immune cells [37].

### 2.2. Cell Surface Marker Expression by Flow Cytometry

Immune cells express several cell surface markers, which each play a different role. Markers expressed on monocytes/macrophages were evaluated on differentiated U937 cells to determine whether vitamin B6 modulates the inflammatory response induced by LPS. U937 cells were seeded at a density of 1 × 10^6^ cells/mL in tissue culture flasks and stimulated with 100 nM VitD_3_ for a total of 72 h (h). Differentiated U937 cells (monocyte/macrophage cells) were stimulated with 1 mg/mL LPS ± 250 μg/mL vitamin B6 for 24 h. Differentiated monocyte/macrophage cells were used for cell surface marker expression and supernatants were collected for cytokine secretion analysis using Bio-Plex (Bio-Rad, Melbourne, VIC, Australia). Flow cytometry was used to determine the expression of cell surface markers, cluster differentiation *(CD)11b*, *CD14*, CD16, *CD86*, CD206, and *MHC-I*. U bottom 96-well plates were seeded with 100 mL of treated cells and controls at 5 × 10^5^ cells per well and incubated with Fc block (1:100 dilution) for 30 min on ice. After washing, cells were labelled with cell surface antibody cocktails and isotype controls linked to fluorochrome, and then incubated on ice for 30 min in the dark. The antibodies were diluted in FACS buffer at the following dilutions according to the manufacturer’s recommendations (*CD11b-PE* 1:400; CD14-BV421 1:200; *CD86-Alexafluor 488* 1:400; *CD16-PE* 1:400; *CD206-PE/Cy7* 1:200; *MHCI-BV510* 1:200) (Bio-Legend, Wangara, WA, Australia, and BD Life Sciences Victoria Australia, Macquarie Park, NSW, Australia). Cells were washed and re-suspended in 300 mL FACS buffer and transferred to FACS tubes. Cells were collected using the BD FACS Canto II, using the Cell Quest program (BD, Victoria Australia), and % cell surface marker expression was analysed using BD FACS Diva software.

### 2.3. Bio-Plex Cytokine Assay

The Bio-Plex human cytokine immunoassay is a highly sensitive and reproducible magnetic bead-based assay that allows accurate measurement of low levels of human cytokines. The Bio-Plex cytokine assay uses 8 mm magnetic beads coated with antibodies against an array of cytokines. Cytokine assays were performed using a bead-based multiplex immunoassay (MIA, 9 Bio-Plex Panel B, Bio-Rad Laboratories Inc. Melbourne, VIC, Australia) that included the cytokines *IL-1β*, *IL-2*, *IL-4*, *IL-6*, *IL-8*, *IL-10*, *IL-17α*, *IFN-γ*, and *TNF-α*. U937 cells were seeded at an appropriate density in tissue culture flasks and stimulated with 100 nm VitD_3_ for a total of 72 h. Differentiated U937 cells were then stimulated with (A) 1 μg/mL LPS, (B) 1 μg/mL LPS + 250 μg/mL vitamin B6, or (C) untreated control. A, B, and C were performed in triplicate. Cells were cultured for 1 h, 3 h, 6 h, 12 h, 24 h, 72 h, and 144 h to assess the time-course expression of cytokines. Supernatants were collected at the end of each time period and placed immediately at −80 °C to prevent cytokine degradation. Standard low photomultiplier tube settings were prepared with “blank” negative controls in duplicate. Ninety-six-well plates were coated with beads, followed by the addition of samples and standards and detection antibodies, and then streptavidin-phytoerythrin as per the manufacturer’s instructions. The beads were re-suspended, and the fluorescence output was read and calculated on the Bio-Plex array reader (Bio-Rad, Melbourne, VIC, Australia). Statistical analysis of data included the mean and standard deviation (SD), as well as a Two-Way ANOVA, followed by Sidak’s multiple comparison test using GraphPad Prism (GraphPad Software version 8, Boston, MA, USA). Significance was defined as *p ≤* 0.05.

### 2.4. RT^2^ Profiler PCR Array for Human Innate and Adaptive Immune Responses

The RT^2^ Profiler polymerase chain reaction (PCR) array contains 84 genes related to innate and adaptive immunity and was purchased from QIAGEN; the kit includes the relevant primers in each well. This kit was used for pathway expression profiling to assess the role of vitamin B6 in modulating monocyte/macrophage inflammatory responses to that induced by LPS. LPS is widely used as a powerful activator of monocytes and macrophages and induces the production of key inflammatory mediators [37]. The human innate and adaptive immune responses PCR array consists of 84 genes relating to IL-1R and Toll-like receptor (TLR) signalling pathways involved in pathogen detection and host defence against bacteria, acute phase response, complement activation inflammatory response, and antibacterial humoral response, as well as genes involved in innate immune response and septic shock. U937 cells were seeded at an appropriate density in tissue culture flasks and differentiated with 100 nm VitD_3_ for 72 h. Differentiated U937 cells (monocyte/macrophage cells) were stimulated with either (A) 1 μg/mL LPS or (B) 1 μg/mL LPS + 250 μg/mL vitamin B6. Both A and B were performed in triplicates. Cells were cultured for 24 h and both adherent and non-adherent cells were removed from the flask, washed, and the pellet was snap-frozen in liquid nitrogen and stored immediately at −80 °C until used.

### 2.5. RNA Extraction from Cells

Extraction of RNA from vitamin B6 cultured cells was performed using an RNeasy1 mini kit (Qiagen, Hilden, Germany) as per the manufacturer’s instructions. Messenger RNA (mRNA) from treated cells was extracted from each cell pellet by disrupting cells with lysis buffer/mercaptoethanol mix. Cells were lysed, and the lysate was placed in the supplied Qia-shredder columns for homogenization, and then transferred to RNeasy mini-spin columns. DNase was used to eliminate any genomic DNA contamination with the RNase-free DNase set (Qiagen, Hilden, Germany). Samples were evaluated for RNA integrity number (RIN) using the Agilent 2100 Bioanalyser and Agilent RNA 6000 nano kit (Agilent Technologies, Santa Clara, CA, USA). All samples tested and used had an RIN higher than the cut-off point of 8.

### 2.6. Assessing Change in Gene Expression

RNA was reverse transcribed using an RT2 first strand kit (Qiagen, Hilden, Germany) and the resultant cDNA was used on the real-time RT2 Profiler PCR Array for innate and adaptive immunity (QIAGEN, Cat. no. PAHS-052Z), in combination with RT2 SYBR^®^ Green qPCR Mastermix (Cat. no. 330529) for the evaluation of gene expression. LPS alone and LPS with 250 μg/mL vitamin B6 were analysed in comparison to untreated (no LPS or vitamin B6) cells using the Qiagen web-based software for the calculation of fold change and results and compared using criteria of >2.0-fold increase/decrease in gene expression as biologically meaningful.

### 2.7. Analysis of Data

Fold-Change (2^ (-Delta Delta CT)) is the normalised gene expression (2^ (-Delta CT)) in the test samples (LPs alone and LPS + vitamin B6) divided by the normalised gene expression (2^ (-Delta C)) in the control sample (Untreated). CT values for each group were exported into Excel to create a Table of CT values, which was then uploaded to the data analysis web portal (http://www.qiagen.com/geneglobe (accessed on 5 April 2021)). Samples were categorised into control and test groups. CT values were normalised based on the manual selection of reference genes. Reference genes are endogenous genes employed as a reference for certifying the authenticity of RT-PCR results. The basis for this internal referencing presumes that the less variation in endogenous gene expression, the better the experimental outcome [38]. The data analysis web portal calculates fold change/regulation using the delta-delta CT method, in which delta CT is calculated between the gene of interest (GOI) and an average of the reference genes (HKG), followed by delta-delta CT calculations (delta CT (Test Group)-delta CT (Control Group)). Fold change is then calculated using the 2^ (delta-delta CT) formula. Fold change data were graphed using prism software whereby fold change in the two treatment groups (LPS and B6 + LPS) was plotted against the control group (untreated) represented as 0 on the *x*-axis. Further analysis was performed using LPS as a control, and the difference in fold change between B6 + LPS and LPS was plotted on the same graphs along with relevant *p* values between the two conditions.

### 2.8. Statistical Analysis

Calculations of *p* values was performed using Student’s *t*-test of the triplicate 2^ (-Delta CT) [(2^-ΔCT)] values for each gene in the control group versus the treatment groups. *p* values less than 0.05 were considered significant. The *p*-value calculation used was based on parametric, unpaired, two-sample equal variance, and two-tailed distribution. Genes were considered significantly altered and included in the analysis if they displayed a greater than >2.0-fold change up/down and a *p*-value of *p* ≤ 0.05.

## 3. Results

LPS is the main component of the outer membrane of Gram-negative bacteria and acts as a powerful activator of cells concerning cytokines, chemokines, and cell surface markers, making them pro-inflammatory. LPS is widely used to study inflammation and was used in all experiments to induce an inflammatory profile to determine the effects of vitamin B6 in an inflammatory environment.

### 3.1. Vitamin B6 Changes Cell Surface Marker Expression in LPS-Stimulated Monocytes

Monocytes and macrophages express cell surface markers that determine their activation state. *CD14* is expressed by monocytes, macrophages, and dendritic cells, and acts as a co-receptor (together with TLR-4) for detection and activation via bacterial LPS. *CD40*, *CD80*, and *CD86* are primarily involved in determining the activation state of mature antigen-presenting cells such as dendritic cells, monocytes, and macrophages. Expressions of *CD40*, *CD80*, or *CD86* on mature antigen-presenting cells is required for the stimulation of T cells. *CD206* (or mannose receptor), a pathogen recognition receptor that binds microbial antigens, is expressed on macrophages and immature dendritic cells. *CD206* is considered a marker of the M2 macrophage phenotype [39]. *CD209* activates phagocytosis in macrophages by binding to pathogen-associated molecular patterns (PAMPS) found in bacteria, viruses, and fungi [40]. Expression levels of *CD14*, *CD40*, *CD80*, *CD86*, *CD206*, and *CD209* on differentiated U937 monocyte/macrophages were determined. LPS was used to induce an inflammatory profile and vitamin B6 was added to determine its effects on cell surface markers.

An increase or decrease change of 2–2.5-fold was considered to be of interest when determining the effect of vitamin B6 + LPS compared to LPS alone. In this regard, *CD14* was increased from 26.5% to 55.5% in the presence of LPS, which was further increased to 70% in LPS + vitamin B6, suggesting that B6 synergistically increases the expression of *CD14* stimulating the cells with LPS (Figure 1). Similarly, *CD40*, *CD80*, and *CD86* were increased in the presence of LPS and were further increased in LPS + vitamin B6, indicating that vitamin B6 stimulates monocyte/macrophage cells; there were no major differences in the expression of *CD206* and *CD209*.

### 3.2. Vitamin B6 Decreases Secretion of IL-1β, IL-6, IL-10, and TNF-α in LPS-Stimulated Monocytes

A time-course Bio-Plex immunoassay was used to study the effects of vitamin B6 on LPS-stimulated monocyte/macrophage cells. LPS was shown to induce an activated inflammatory profile by stimulating the secretion of *IL-6*, *IL-10*, *TNF-α*, and *IL-1β* compared to the non-stimulated control (Figure 2). Two-way ANOVA and Sidak’s multiple comparison tests were used to analyse data for *p*-value significance, using Prism graph pad software. The addition of vitamin B6 to LPS-stimulated cells significantly decreased the expression of *IL-6* at 12 h and 72 h (*p* ≤ 0.05), *TNF-α* at 6 h (*p* ≤ 0.00005) and 12 h (*p* ≤ 0.005), *IL-10* at 12 h (*p* ≤ 0.005) and 24 h (*p* ≤ 0.05), and *IL-1β* at 72 h (*p* ≤ 0.00005) and 144 h (*p* ≤ 0.05). Vitamin B6 did not alter the secretion of other cytokines (*IL-2*, *IL-4*, *IL-8*, *IL-17α*, and *IFN-γ*) measured.

### 3.3. Vitamin B6 Decreases Inflammatory Gene Expression in LPS-Stimulated Monocyte Cells

Pathway expression profiling, using RT^2^ profiler PCR arrays for human innate and adaptive immune responses, was used to assess the effects of LPS- ± vitamin B6-stimulated monocyte/macrophage cells against non-stimulated control cells. Vitamin B6 showed clear modulation by downregulating 83 genes that LPS had upregulated compared to untreated control; 54 of these genes showed a fold change greater than two, which were deemed biologically relevant. Significance between the two conditions (LPS) and (vitamin B6 + LPS) and control (untreated) is indicated by *p* values (*p* ≤ 0.05). The difference in fold change between (B6 + LPS) and (LPS) is plotted on the same graphs along with relevant *p* values between the two conditions (Figure 3).

### 3.4. Vitamin B6 Downregulates the Expression of Cytokine Genes

Cytokines are secreted by monocyte/macrophage cells in both the innate and adaptive immune systems as a response to pathogenic infection. Cytokines released by innate immune cells aid the front line of defence by controlling opportunistic invasion from microbial pathogens including bacteria and viruses. Within the adaptive immune system, cytokines mediate growth and differentiation and activate different types of effector cells for microbial elimination. Cytokines enable cells to regulate immune response, and pro-inflammatory cytokines activate immune cells and instigate the release of additional cytokines. Chemokines are a group of chemoattractant cytokines that function to induce chemotaxis in proximal responsive cells. This study shows that vitamin B6 can significantly modulate cytokine responses in LPS-induced monocyte/macrophage cells.

Chemokine Ligand-5 (*CCL5*/RANTES) functions as a ligand for chemokine receptors *CCR1*, *CCR3*, and *CCR5* and is a chemotactic cytokine for macrophages and T cells. *CCL5* also facilitates the recruitment of leukocytes to sites of inflammation [41,42]. *CCL5* was downregulated by vitamin B6 2.8-fold from 3.8-fold (LPS) to 1-fold (B6 + LPS) (*p* ≤ 0.005) (Figure 4A).

*IL-18*, an IFN-gamma-inducing factor (IGIF), is a strong inducer of inflammatory cytokines including IFN-gamma, and its biological activity is mediated via its binding to the *IL-18* receptor complex and activation of the *NF-κβ* pathway [43,44]. Vitamin B6 downregulated *IL-18* expression by 3-fold from 1.4-fold (LPS) to -1.5-fold (B6 + LPS) (*p* ≤ 0.05) (Figure 4A). *IL-5* is frequently associated with eosinophils and asthma [45,46] but was also shown in some animal studies to polarise monocytes towards an anti-inflammatory phenotype [47]. *IL-5* expression was downregulated by vitamin B6, 3.6-fold from 1.4-fold (LPS) to -2.2-fold (B6 + LPS) (*p* ≤ 0.05) (Figure 4A).

The colony-stimulating factor 2 (*CSF2*) gene encodes the cytokine *GMSF-2* (Granulocyte-macrophage colony-stimulating factor). *GMSF-2* is expressed by myeloid cells and controls the production, differentiation, and function of granulocytes and macrophages, and plays a role in tissue inflammation [48]. LPS + B6 v LPS alone reduces *CSF2* expression 6.8-fold from 18.1-fold (LPS) to 11.3-fold (B6 + LPS) (*p* ≤ 0.05) (Figure 4B).

Interleukin 10 (*IL-10*) is secreted by monocytes/macrophage cells (and other immune cells in certain contexts) [49]. Its ability to limit the host’s immune response to pathogens renders it a potent anti-inflammatory cytokine. *IL-10* produced by monocytes to alternatively activate macrophages *IL-10* can also downregulate the expression of other pro-inflammatory cytokines produced by activated macrophages such as *TNF-α*, *IL-6*, and *IL-1* [50]. LPS upregulated the expression of *IL-10* 9.2-fold, which was reduced to 3.9-fold in the presence of vitamin B6, a change in expression of 5.3-fold (*p* ≤ 0.05) (Figure 4B).

C-C Motif Chemokine Ligand 2 (*CCL2*) is a chemokine ligand also referred to as monocyte chemoattractant protein-1 (*MCP-1*). It is a chemotactic protein for myeloid and lymphoid cells and reacts synergistically with other inflammatory stimuli to pathological and physiological circumstances during immune defence; it is responsible for the recruitment, regulation, and polarisation of macrophages during inflammation [51]. *CCL2* was downregulated by vitamin B6 5-fold from 8.3 (LPS) to 3.3 (B6 + LPS) (*p* ≤ 0.05) (Figure 4B).

C-X-C Motif Chemokine Ligand 10 (*CXCL10*) (IP-10) encodes an anti-microbial chemokine that stimulates monocytes upon binding of the receptor *CXCR3*. The *CXCR3* receptor also reacts with the chemokines *CXCL9* (MIG), *CXCL11* (I-TAC/IP-9), and *CXCL4*, all of which function as immune chemo-attractants associated with interferon-induced inflammatory responses [52]. *CXCL10* was downregulated by vitamin B6 34-fold from 38.8 (LPS) to 4.7-fold (B6 + LPS) (*p* ≤ 0.05) (Figure 4C). *CXCL8* (IL-8) is a chemoattractant cytokine, released in response to inflammation by monocytes and macrophages, targeting neutrophils, basophils, and T cells involved in inflammation and chemotaxis [53]. It is considered a major mediator of inflammatory response and is rapidly produced in response to bacterial and viral infiltration. LPS + B6 vs. LPS reduced gene expression by 25-fold from 69-fold (LPS) to 44.4-fold (B6 + LPS) (*p* ≤ 0.05) (Figure 4C).

### 3.5. Vitamin B6 Downregulates Expression of Inflammatory and Defence Response Genes

The regulation and control of cellular defence in humans is vital for protection from environmental stresses, ranging from microbial infections to ageing, diet, and lifestyle choices. Defence response involves the activation of specific pathways that utilise cellular machinery to create physiological and behavioural cell responses, often by triggering the transcriptional activation of inflammatory gene sets [54]. Inflammatory response to harmful stimuli such as LPS initiates the transcriptional activation of a variety of genes aimed at maintaining host defence and developing acquired immunity [55].

Pattern recognition receptor (PPR) genes encode proteins for pattern recognition receptors, which are part of the innate immune system and play a pivotal role in their ability to recognise molecules that are frequently found in pathogens. Pathogen-associated Molecular-Patterns (PAMPs) are commonly found in pathogens such as viruses or bacteria but not in eukaryotic organisms [56]. Damage-associated Molecular-Patterns (DAMPs) are molecules released by damaged cells. LPS is an example of a PAMP and the cell’s response to LPS is a microbiocidal and pro-inflammatory reaction due to the signal engagement of PRRs [56]. PRRs are divided into four main subgroups. Toll-like receptors (TLRs) extracellular signalling receptors [57,58], C-type lectin receptors (CLRs); extracellular receptors, NLR or nucleotide-binding oligomerisation domain-like receptors (NODs); intracellular receptors, RIG-1-like receptors (RLRs); and cytoplasmic receptors. Of these, both TLR and NLR recognise LPS [56]. Gene array results indicated that vitamin B6 was able to downregulate the pro-inflammatory effect of LPS on PRR genes.

NOD-LRR- and pyrin domain-containing protein 3 (*NLRP3*) is an intracellular sensor for microbial detection, which forms the *NLRP3* inflammasome triggering the release of inflammatory cytokines via the caspase-1-dependent pathway [59]. Vitamin B6 decreased the expression of *NLRP3* 3.8-fold from 1.4 (LPS) to −2.5 (LPs + B6) (*p* ≤ 0.005) (Figure 5A). DExD/H-Box Helicase 58 (*DDX58*) encodes for the protein RIG-1, an antiviral innate immune response receptor, that, in response to viral nucleic acids, initiates a pro-inflammatory signalling cascade. This gene may also stimulate the production of granulocytes, bacterial phagocytosis, cell migration regulation, and cellular differentiation [60]. *DDX58* expression was downregulated by vitamin B6 2.5-fold from 3.6-fold (LPS) to 1.1 (B6 + LPS) (*p* ≤ 0.005) (Figure 5).

Nucleotide-binding Oligomerisation Domain-like receptors (*NOD1* and NOD 2) are intracellular receptors that specialise in detecting microbial pathogens with the ability to invade and multiply intracellularly. Once these receptors are activated by pathogens, they activate signalling pathways, which trigger transcriptional responses leading to the expression of a pro-inflammatory gene set [61]. Gene array results show that vitamin B6 downregulates the pro-inflammatory effect of LPS on bacterial defence genes *NOD1* 3.9-fold from 1-fold (LPS) to -2.5-fold (B6 + LPS) (*p* ≤ 0.005) and *NOD2*, by 1.9-fold from 3.2-fold (LPS) to 2.4-fold (B6 + LPS) (Figure 5A).

Toll-like receptors (TLRs) have emerged as an important part of the antimicrobial host defence response within the adaptive immune system as they recognise PAMPS from numerous microbes and are responsible for the activation of transcription factors *NF-κβ* and IRFs [62]. Cell surface TLRs are responsible for the recognition of microbial membrane components such as proteins, lipoproteins, and lipids [57,58]. These include TLR1, TLR2, TLR4, TLR5, TLR6, and TLR10. Intracellular TLRs recognise bacterial and viral nucleic acids and self-nucleic acids in autoimmunity; these include TLR3, TLR7, TLR8, TLR11, and TLR12. TLR-4 recognises LPS and uses both *MYD88*- and TRIF-dependent signalling pathways and selectively recruits TRAM adapter protein [63]. Vitamin B6 downregulated *TLR4* expression by 3.2-fold from 1.5-fold (LPS) to -1.75-fold (B6 + LPS) (*p* ≤ 0.05) (Figure 5A). TLR-9 binds bacterial DNA and activates *NF-κβ* via the *MYD88* and TRAF6 pathways. Vitamin B6 downregulated *TLR9* expression by 4.2-fold from 1.2 (LPS) to -2.8-fold (B6 + LPS) (*p* ≤ 0.05) (Figure 5A). The following TLR genes were also downregulated by vitamin B6; *TLR1* was down by -3-fold from 9.4-fold (LPS) to 6.4-fold (B6 + LPS) (*p* ≤ 0.05), *TLR2* was down by 4.6-fold from 8.6 (LPS) to 4-fold (B6 + LPS) (*p* ≤ 0.05), *TLR5*—3.7-fold from 1.1 (LPS) to -2.7 (B6 + LPS) (*p* ≤ 0.05), *TLR7*—4.8 fold from 5.9-fold (LPS) to 1.1-fold (B6 + LPS) (*p* ≤ 0.0005), *TLR8*—4.4 fold from 5.5-fold (LPS) to 1.1-fold (B6 + LPS) (*p* ≤ 0.0005) (Figure 5A).

The *MYD88* gene provides instructions for making a protein (myeloid differentiation primary response protein 88) involved in cross-membrane signalling within immune cells. *MYD88* is used by all TLRs, it responds to inflammatory response via cytokine secretion and activates *NF-κβ* via IRAK1, IRAK2, IRF7, and TRAF6 signalling pathways, as well as activating the MAPK signalling pathway [64]. The presence of vitamin B6 in monocyte/macrophage LPS-stimulated cells was shown to downregulate *MYD88* expression by 3.2-fold from 1.4-fold (LPS) to -1.8-fold (*p* ≤ 0.05) (Figure 5D).

Cytokines, IL-1-beta (*IL-1β*), and IL-1-alpha (IL-1α) are equally potent pro-inflammatory cytokines and are produced in response to inflammation caused by infections and microbial endotoxins and other inflammatory agents. *IL-1β* is known to be responsible for contributing to further damage during chronic disease [65] and is implicated in many inflammatory conditions such as sepsis, inflammatory bowel disease, and rheumatoid arthritis. LPS + B6 compared to LPS alone reduced *IL1β* expression by 97-fold from 219.5-fold (LPS) to 121.8-fold (*p* ≤ 0.05) (Figure 5B). *IL-6* is known to respond with broad-ranging effects to infection and tissue injury, and contribute to the host’s defence [66]. *IL-6* has pleiotropic effects, as it is both anti- and pro-inflammatory [46,67], and works both to inhibit Th1 polarisation and promote Th2 differentiation [68]. Excessive *IL-6* synthesis is implicated in several disease pathologies including COVID-19 [66,69]. Vitamin B6 significantly downregulated *IL6* by 17.8-fold from 21.6-fold (LPS) to 3.8-fold (B6 + LPS) (*p* ≤ 0.005) (Figure 5C). *IL-23-a* is produced by macrophages and dendritic cells and is an important part of the inflammatory response in peripheral tissues. *IL-23-a* expression is substantially increased in several human cancers [70], leading to its tumour-promoting properties. In addition, IL-23-producing macrophages are involved in inflammatory responses [71]. The addition of vitamin B6 in LPS stimulated macrophage/monocyte cells significantly downregulated *IL23A* expression by 3.4-fold from 1.9-fold (LPS) to -1.5-fold (*p* ≤ 0.05) (Figure 5D), suggesting vitamin B6 to be anti-inflammatory. Tumour necrosis factor-alpha (*TNF-α*) is an inflammatory cytokine secreted mainly by macrophages, which assist in the regulation of proliferation and differentiation, as well as apoptosis, coagulation, and lipid metabolism [72]. LPS + B6 v LPS alone reduce *TNF* expression by 1.6-fold from 2.8-fold (LPS) to 1.2-fold (B6 + LPS) (*p* ≤ 0.005) (Figure 5D). Vitamin B6 is anti-inflammatory by decreasing *IL-1*, *IL-6*, *IL-23*, and *TNF-α* expression in monocyte/macrophage cells.

*LYZ* is a gene encoding human lysozyme that acts as an antibacterial enzymatic agent and is found in leucocytes within human milk, spleen, lungs, plasma, saliva, and tears. Lysozyme is considered one of the most vital anti-bacterial agents in human and animal immunity. A recent study has demonstrated the anti-inflammatory action of lysozyme via gene regulation involving the *TNF-α*/*IL-1β* pathways in monocytes [73,74]. Vitamin B6 was shown to downregulate *LYZ* expression by 4.9-fold from 1.1-fold (LPS) to -3.8-fold (*p* ≤ 0.0005) (Figure 5D).

Complement component 3 (*C3*) is a key activator of the complement system, involving both classical and alternative complement pathways. Pathogenic invasion triggers the cleaving of the *C3* protein into two segments, *C3a* and *C3b*. *C3a*, also known as *C3a* anaphylatoxin, is a modulator of inflammation and demonstrates antimicrobial activity. *C3b* acts as a regulating protein in complement system response [75,76]. Vitamin B6 was shown to downregulate *C3* expression by 5.4-fold from 8.1-fold (LPS) to -1.3-fold (*p* ≤ 0.005) (Figure 5D).

Forkhead box protein P3 (*FOXP3*), is a transcription factor that regulates immune control and is crucial in aiding the regulatory activity of regulatory T cells (Tregs). It is responsible for much of the cell’s ability to suppress immune function [77]. *FOXP3* was also shown to have a suppressive function and promote tumour growth [78]. *FOXP3*+ macrophages can inhibit *CD4*+ T cells, and can inhibit *FOXP3* expression via small interfering RNA knockout to abrogate their T cell suppressive ability [78]. Vitamin B6 was shown to downregulate *FOXP3* expression in monocyte/macrophage cells by 2.9-fold from 1.2-fold (LPS) to -1.7-fold (*p* ≤ 0.05) (Figure 5D), suggesting that vitamin B6 may restore T cell functionality in vivo, as it decreases *FOXP3* expression in monocyte/macrophage cells.

Signal transducer and activator of transcription 3 (*STAT3*) is important for its role in the maturation of immune cells, and its immune response to bacteria and fungi [79]. High levels of STAT-3 in the tumour microenvironment, especially expressed by monocyte/macrophage cells, result in a poorer prognosis, i.e., promote an immunosuppressive phenotype [80]. Herein, it was noted that *STAT-3* expression was downregulated by vitamin B6 by 3.2-fold from 1.9-fold (LPS) to -1.2-fold (B6 + LPS) (*p* ≤ 0.005) (Figure 5D), indicating that this lowered expression would promote anti-cancer properties in vivo.

### 3.6. Vitamin B6 Downregulates the Expression of other Genes in the Innate Immune System

The caspase 1 (Casp1) gene encodes a protein of the caspase (Cysteine-aspartic acid protease) family. Caspase 1 plays a role in innate immunity by responding to cytosolic signalling via various inflammasomes, and initiates a two-fold response. The first deals with the activation and secretion of the pro-inflammatory cytokines *IL-1β* and *IL-18*, and the second response involves the triggering of pyroptosis, a form of lytic cell death forming part of the antimicrobial response [81]. Vitamin B6 was shown to downregulate *Casp1* expression by 3.5-fold from 2.2 (LPS) to -1.3-fold (*p* ≤ 0.005) (Figure 6A).

Cluster differentiation 4 (*CD4*) molecule is expressed by T helper cells, but also by monocytes and macrophages. It is a membrane glycoprotein that performs an essential role in the immune response by assisting the T cell receptor (TCR) and amplifying TCR signalling. *CD4* directly interacts with MHC class II molecules on antigen-presenting cells and is a primary receptor for the entry of HIV into host cells [82]. Vitamin B6 was shown to downregulate *CD4* expression by 3.5-fold from 2.1-fold (LPS) to -1.5-fold (*p* ≤ 0.05) (Figure 6A).

Major Histocompatibility complex class E (*HLA-E*) molecules are present in both innate and adaptive immune function, modulating either activation or inhibition of natural killer (NK) cytotoxicity and cytokine production. *HLA-E* also plays a part in viral or bacterial peptide presentation, provoking T-cell response [83]. In monocytes, *HLA-E* is upregulated during monocyte–macrophage differentiation [83]. Vitamin B6 was shown to downregulate *HLA-E* expression by 3-fold from 1.8-fold (LPS) to -1.2-fold (*p* ≤ 0.05) (Figure 6A).

Mitogen-Activated Protein Kinase 1 (*MAPK1*/ERK2) and MAPK3/ERK1 are prototypic MAP kinases that function primarily in mitogen-activated signal transduction pathways [84]. They both play an important role in the MAPK/ERK cascade. The protein encoded by the *MAPK1* gene is a component of the MAP kinase family. MAP kinases or Extracellular signal-regulated kinases (ERKs) are diverse in function, mediating processes such as cell differentiation, growth, and adhesion. They also play a role in the regulation of transcription, translation, and arrangement of the cytoskeleton [85]. Vitamin B6 was shown to downregulate *MAPK1* expression by 3-fold from 1.5-fold (LPS) to -1.4-fold (*p* ≤ 0.05) (Figure 6A).

Myeloperoxidase (MPO) is a heme-containing peroxidase that is synthesised and expressed mainly by myeloid cells during myeloid differentiation. It is mostly found in neutrophils and monocytes, released into the extracellular fluid during inflammation [86]. Vitamin B6 was shown to downregulate *MPO* expression by 3.1-fold from 1.5-fold (LPS) to -1.6-fold (*p* ≤ 0.05) (Figure 6A). MX Dynamin-Like GTPase 1 (MX1) gene encodes a guanosine triphosphate (GTP) metabolising protein, induced by type I and type II interferons, which mounts an antiviral response against a variety of RNA and some DNA viruses. Vitamin B6 was shown to downregulate *MX1* expression by 3-fold from 1.4-fold (LPS) to -1.6-fold (*p* ≤ 0.005) (Figure 6A).

Nuclear Factor Kappa B Subunit 1 (*NF-κβ1*) plays an important role in the regulation of immune response to infections and stimuli such as pro-inflammatory cytokines, chemokines, adhesion molecules, and enzymes. *NF-κβ1* is involved in biological processes, which include inflammation, immunity, differentiation, cell growth, tumorigenesis, and apoptosis [87]. Vitamin B6 was shown to downregulate *NF-κβ1* expression by 3-fold from 1.9-fold (LPS) to -1.3-fold (*p* ≤ 0.005) (Figure 6A). *NF-κβ* inhibitor alpha gene (*NF-κβIA*) encodes for the *NF-κβIA* protein. This protein enables *NF-κβ* to remain bound within a protein complex termed IKK. *NF-κβIA* responds to signalling, which allows *NF-κβ* to be released from this complex and move into the nucleus to bind to DNA. *NF-κβ* is responsible for the regulation of many genes that mediate inflammatory responses. One study showed that *NF-κβIA* mutation causes defective *NF-κβ* signalling, leading to hyper *IL-1β* secretion in macrophages [88]. Vitamin B6 was shown to significantly downregulate *NF-κβIA* expression by 4.6-fold from 7.9-fold (LPS) to 3.3-fold (*p* ≤ 0.05) (Figure 6B).

Signal Transducer and Activator of Transcription 1 (*STAT1*) encodes a protein that drives multiple, complex, and contrasting transcriptional functions. It is involved in the inhibition of the IL-17 inflammatory pathway [89], the promotion of interferon-alpha/beta signalling pathways involved in viral defences and interferon-gamma signalling pathways important for bacterial defence [89,90]. Vitamin B6 was shown to downregulate *STAT1* expression by 3.1-fold from 1.9-fold (LPS) to -1.3-fold (*p* ≤ 0.05) (Figure 6A). STAT6 is predominantly stimulated by *IL-4* and *IL-13* and is involved in allergic inflammatory disease [91]. It is known to drive M2 macrophage polarisation [92]. *STAT6* expression was downregulated by vitamin B6 by 3.4-fold from 2.3-fold (LPS) to -1.1-fold (B6 + LPS) (*p* ≤ 0.05) (Figure 6A).

The cell surface receptor *CD14* forms a multi-receptor complex along with TLR-4 and LYS96, which plays a significant role in LPS recognition and innate immune response. Monocytes and macrophages strongly express *CD14*, whilst promonocytes and monoblasts show only weak expression; hence, *CD14* is often viewed as a monocyte differentiation marker [93]. Vitamin B6 was shown to significantly downregulate *CD14* gene expression by 3.5-fold from 12.4-fold (LPS) to 2.7-fold (*p* ≤ 0.005) (Figure 6B). Additionally, *CD80* and *CD86* cell surface markers expressed by activated monocytes/macrophages have a differential role in the regulation of inflammation, especially in the innate immune response to sepsis. The upregulation of *CD80* and loss of *CD86* expression correlates with increased illness severity and inflammation in humans [94]. The decreased expression of *CD86* is also involved in decreased T cell stimulation. *CD80* expression was downregulated by vitamin B6 by 3.2-fold from 8.3-fold (LPS) to 5.1-fold (B6 + LPS), although this change was not significant *p* = 0.37. *CD86* expression, however, was significantly downregulated by vitamin B6, by 6.1-fold from 10.9-fold (LPS) to 4.8-fold (B6 + LPS) (*p* ≤ 0.05) (Figure 6B).

C-C motif chemokine receptor 4 and C-C motif chemokine receptor 5 (*CCR4*, CD194) and (*CCR5*, RANTES) are G protein-coupled chemokine receptors. Chemokines comprise a group of molecules that play a fundamental role in the regulation of leukocyte trafficking, as well as the development and homeostasis of the immune system. *CCR4* regulates the function of inflammatory macrophage in many inflammatory disorders including multiple sclerosis; *CCR4* knockout mice show delayed disease progression [95]. *CCR4* expression on monocyte/macrophage cells was upregulated by 1.1-fold in the presence of LPS but was downregulated by 4.7-fold (*p* ≤ 0.05) in LPS + vitamin B6 cultures. In addition, vitamin B6 significantly downregulated *CCR5* (CD195) expression by 7.6-fold (*p* ≤ 0.005) (Figure 6C). The expression of *CCR5* on monocyte/macrophage cells is involved in inflammatory responses to infection, hence, its downregulation in the presence of vitamin B6 suggests its anti-inflammatory properties. Furthermore, intracellular adhesion molecule (ICAM) is greatly increased in immune cells as a response to inflammatory stimulation. It is best known for its role in directing leukocytes from the circulation to sites of inflammation [96]. Vitamin B6 was shown to downregulate *ICAM* expression by 3.5-fold from 5.3 (LPS) to 1.8-fold (*p* ≤ 0.005) (Figure 6C). Moreover, C-X-X Motif Chemokine Receptor 3 (*CCXCR3*, GPR9, CD 183) is an interferon-inducible chemokine-receptor expressed on monocytes and other cell types that is involved in cytoskeletal changes, chemotactic migration, and activation of integrin [97]. *CXCR3* expression was downregulated in the presence of vitamin B6 from LPS by 3.4-fold from 1.9-fold (LPS) to -1.4-fold (B6 + LPS) (*p* ≤ 0.05) (Figure 6C).

The signal transducer and activator of transcription 6 (STAT6) is predominantly stimulated by *IL-4* and *IL-13* and is involved in allergic inflammatory disease [91]. *STAT6* expression was downregulated by vitamin B6 by 3.4-fold from 2.3-fold (LPS) to -1.1-fold (B6 + LPS) (*p* ≤ 0.05) (Figure 6C). Interferon Regulatory Factor (*IRF3*) is a transcriptional regulator of type I IFN, involved in response to pathogenic infection in the innate immune system [74]. Vitamin B6 was shown to downregulate *IRF3* expression by 3.8-fold from -1-fold (LPS) to -2.8-fold, (*p* ≤ 0.005) (Figure 6C).

Integrin alpha M (*ITGAM*), also known as CD11b or CR3 (complement receptor 3), is a gene encoding integrin adhesion molecule M. It is important for various adhesive interactions involving monocytes, macrophages, granulocytes, and the phagocytosis of complement-coated particles. *ITGAM* is also involved in leukocyte migration in the presence of CD18. Vitamin B6 was shown to downregulate *ITGAM* expression by 2-fold from 3.1-fold (LPS) to 1.1-fold (*p* ≤ 0.05) (Figure 6C). TLR adaptor molecule 1 (TICAM) is an adapter molecule within the innate immune system that responds to pathogenic invasion. TICAM facilitates protein-to-protein interactions between TLRs, especially TLR3 [98], to mediate *NF-κβ* and IRF (interferon regulatory factor) activation during an antiviral immune response [99]. Vitamin B6 was shown to downregulate *TICAM* expression by 3.9-fold from 2.7-fold (LPS) to −1.1-fold (*p* ≤ 0.05) (Figure 6C).

Mitogen-activated protein kinase 8 (MAPK8) also known as JNK1, is a member of the MAP kinase family. MAP kinases oversee a broad range of cellular processes including proliferation, differentiation, transcription regulation, and development. MAPK8 mediates early gene expression in response to cellular stimuli. MAPK8 is involved in *TNF-α*-induced apoptosis; it is also thought to be involved in the cytochrome c-mediated cell death pathway via ultraviolet radiation-induced apoptosis [100,101,102] Vitamin B6 was shown to downregulate *MAPK8* expression by 3.3-fold from 1.2-fold (LPS) to -2.1-fold (*p* ≤ 0.05) (Figure 6C).

## 4. Discussion

This is the first study to demonstrate the extensive broad array of anti-inflammatory effects of high-dose vitamin B6 on an LPS-activated monocytes/macrophage cell line. Vitamin B6 was shown to downregulate genes associated with inflammatory responses and defence responses to bacteria. These include inflammatory cytokines and chemokines, pattern recognition receptors, and cell surface markers. LPS is a key component of the outer membrane of Gram-negative bacteria and is a potent activator of the highly LPS-sensitive monocytes and macrophages with an ability to upregulate the production of key inflammatory mediators such as cytokines, chemokines, and other inflammatory-related proteins [103,104]. LPS monocyte stimulation induces an array of genes that express inflammatory mediators, and the addition of vitamin B6 to the cell cultures was shown to downregulate these responses. The ability of vitamin B6 to downregulate inflammatory cytokine production was previously reported. Supplementation of vitamin B6 was shown to suppress *TNF-α* and *IL-6* levels in patients with rheumatoid arthritis [105]. Likewise, bone marrow-derived macrophages treated with pyridoxal and stimulated with LPS reduced *IL-1β*, *TNF-α*, and *IL-6* [35]. In another study, vitamin B6 downregulated inflammation (decreased *IL-1β*, *IL-6*, and *TNF-α*), delayed death, and increased the survival rate in high-dose LPS-treated mice compared to the controls [35]. Vitamin B6 was also found to protect mice from toxic effects induced by LPS by preventing *IL-1Β* protein production via the inhibition of *NLRP3* inflammasome activation [106] and also inhibits activation of NF-κB in LPS-stimulated mouse macrophages [107]. Furthermore, high-dose B6 combined with compound amino acid was shown to prevent inflammation by inhibiting the HMNGB1/TLR4/NF-κB signalling pathway [108]. Interestingly, we note that vitamin B6 significantly downregulated *NF-κB IL-6*, *IL-1β*, *TNF-α*, and *NLPR3*, which aligns with these previous studies.

Monocytes are crucial players in the innate immune system and as such, rely on PRRs to detect pathogens and initiate host defence responses. Amongst these are TLRs, which play a vital part in host defence response and activation, direct microbiocidal activity, and activate inflammatory pathways [109,110]. TLRs on the surface of cells recognise microbial membrane components, whilst intracellular TLRs recognise bacterial and viral nucleic acids. TLR4 recognises LPS and utilises *MYD88*- and TRIF (TICAM)-dependent signalling pathways [63,111,112]. Herein, we note that vitamin B6 was also able to downregulate both cell surface and intracellular TLRs as well as *MYD88* and TICAM gene expression. Other genes (*STAT1*, *STAT3*, *C3*, *LYZ*, *L96*, and *CCL5*) relating to the inflammatory response to bacteria were also shown to be downregulated in the presence of vitamin B6.

Although it is interesting to consider the downregulation of inflammatory response gene by gene, the value of this study lies in the overall context of the data. Vitamin B6 appears to be effective in mediating an overall dampening of pro-inflammatory responses induced by LPS stimulation. The importance of these findings lies in the potential therapeutic use of vitamin B6 in the context of hyperinflammatory immune disorders. Gram-negative sepsis is associated with significant morbidity and mortality in both adults and children worldwide [113,114]. Sepsis occurs when the immune system succumbs in the battle against severe infection. Immune cell activation via LPS from the cell wall of infecting bacteria during sepsis can cause the production of inflammatory cytokines such as *IL-6*, *IL-1β*, and *TNF-α* [115]. If the immune system is overwhelmed and dysregulated by bacterial load, a “cytokine storm” can ensue, producing catastrophic inflammation and ultimately leading to organ failure and death [115]. Currently, there are no drugs specifically approved to treat sepsis, although some drugs were (unsuccessfully) trialled that target particular inflammatory pathways, for example, Cytofab, a drug made by AstraZeneca, targeting *TNF-α* [116].

A small percentage of patients who contract the lethal SARS-CoV-2 virus display a similar dysfunctional immune response whereby a ‘cytokine storm’ can lead to acute respiratory distress syndrome and multiple organ failure [117]. These patients also exhibit high levels of inflammatory factors such as *TNF-α*, *IL-1β*, *IL-6*, *IL-10*, *CXCL8*, and *IFN-γ* [118]. It appears that many of these inflammatory pathways that are upregulated in cytokine storm in both patients with sepsis and COVID-19 are the same pathways downregulated by vitamin B6 in LPS-stimulated monocytes. It is possible that the use of a natural immunosuppressant such as high-dose vitamin B6 as a monotherapy, or in combination with other drugs, may be useful in targeting the inflammation in cytokine storms associated with these hyper-inflammatory conditions. High-dose intravenous vitamin C monotherapy was shown to possess some benefits in critically ill sepsis patients by lowering inflammatory markers, as has high-dose vitamin C (up to 10,000 mg/day) in combination with hydrocortisone [119], but to date, vitamin B6 has not been studied for use in humans for the treatment of cytokine storm in sepsis or COVID-19 patients, although one study has shown that high-dose B6 can successfully reduce oxidative stress and exert anti-inflammatory effects in peripheral organs of cecal ligation puncture-induced infection in adult male Wistar rats [120].

Although the potential of these findings is exciting, a note of caution is due here, given the limitations of this study. The main limitation is that monocytes and macrophages were studied in isolation, when in fact the inflammatory microenvironment includes numerous other immune cells and the crosstalk between them could greatly affect the data, we note in vitro, than when studied in vivo. Furthermore, only one dosage and time point were studied, and protein studies carried out in this study over different time points indicate that cytokines have their own timelines, and this may be an important factor in the administration of dosages in therapy situations.

## 5. Conclusions

In conclusion, this study has identified that a high dose of vitamin B6 has a global anti-inflammatory effect on LPS-induced inflammation in monocyte/macrophage cells by downregulating the key broad-spectrum inflammatory mediators *CCL2, CCL5, CXCL2, CXCL8, CXCL10, CCR4, CCR5, CXCR3, IL-1β, IL-5, IL-6, IL-10, IL-18, IL-23-a, TNF-α, CSF2, DDX58, NLRP3, NOD1, NOD2, TLR-1 -2 -4 -5 -7 -8 -9, MYD88, C3, FOXP3, STAT1, STAT3, STAT6, LYZ, CASP-1, CD4, HLA-E, MAPK1, MAPK8 MPO, MX-1, NF-κβ, NF-κβ1A, CD14, CD40, CD40LG, CD86, Ly96, ICAM1, IRF3, ITGAM,* and *IFCAM2*. Although further studies are required to understand whether these effects can be translated into in vivo animal models and amongst cross talk from other immune cells in the complex inflammatory micro-environment, the findings herein show promise regarding vitamin B6 within the context of a potent broad-spectrum anti-inflammatory mediator and could be useful as an adjunct treatment for inflammatory-related diseases.

## Figures and Tables

**Figure 1 biomedicines-11-02578-f001:**
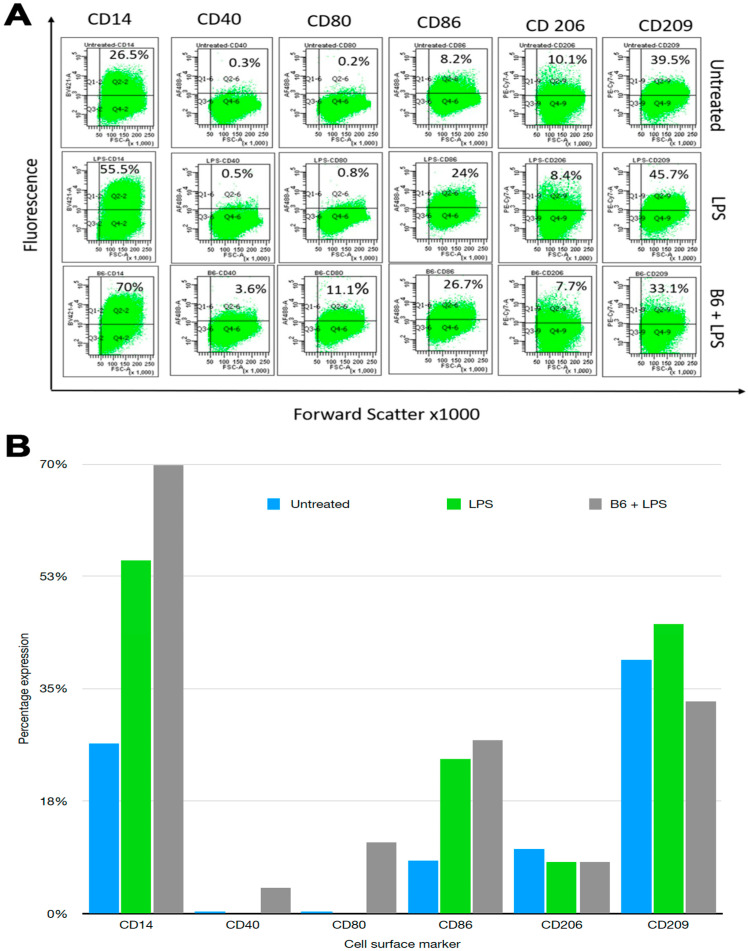
Cell surface marker expression. Percentage expressions are represented as (**A**) dot plots. U937 cells were differentiated into monocyte/macrophage cells, followed by stimulation with LPS + vitamin B6 for 24 h. Quadrants were determined based on isotype controls. Top right quadrant represents cells that are surface marker positive; (**B**) represents bar graphs of the dot plots.

**Figure 2 biomedicines-11-02578-f002:**
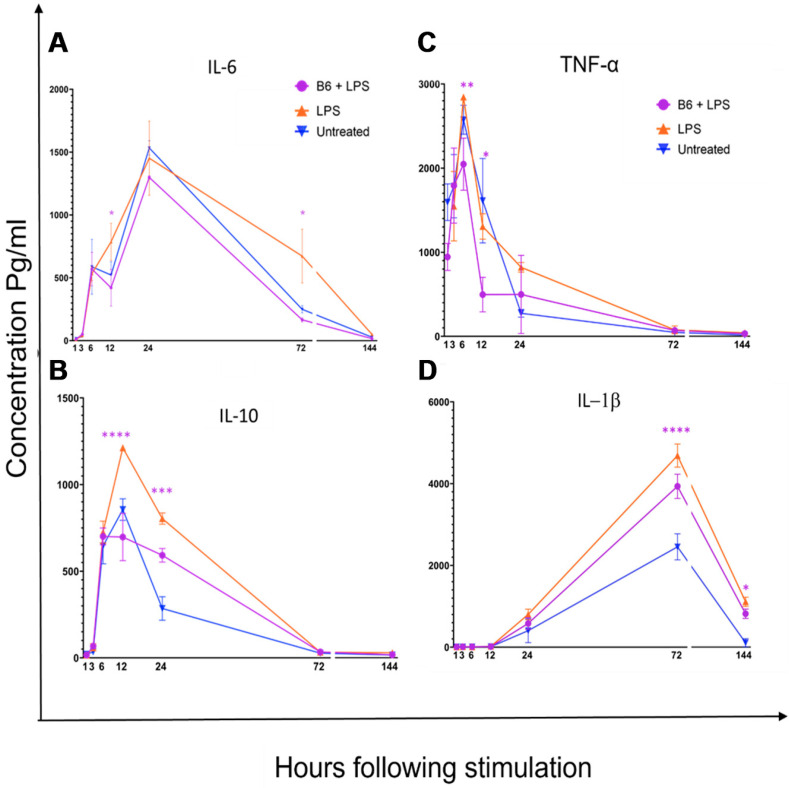
The effect of vitamin B6 on cytokine expression in LPS-stimulated monocytes. U937 cells were differentiated into monocyte cells and incubated with LPS to trigger an inflammatory response (orange lines). Vitamin B6 was added to LPS-stimulated cultures to determine its modulatory effects. Supernatants of the cultures were collected over 1–144 h and (**A**) *IL-6*, (**B**) *IL-10*, (**C**) *IL-1β*, and (**D**) *TNF-α* cytokines were measured using a Bioplex assay kit. Two-way Anova and Sidak’s multiple comparison tests were used to analyse data for *p* value significance, using Prism graph pad software. Significance was demonstrated as * *p* < 0.05, ** *p*< 0.005, *** *p* < 0.0005 **** *p* < 0.00005.

**Figure 3 biomedicines-11-02578-f003:**
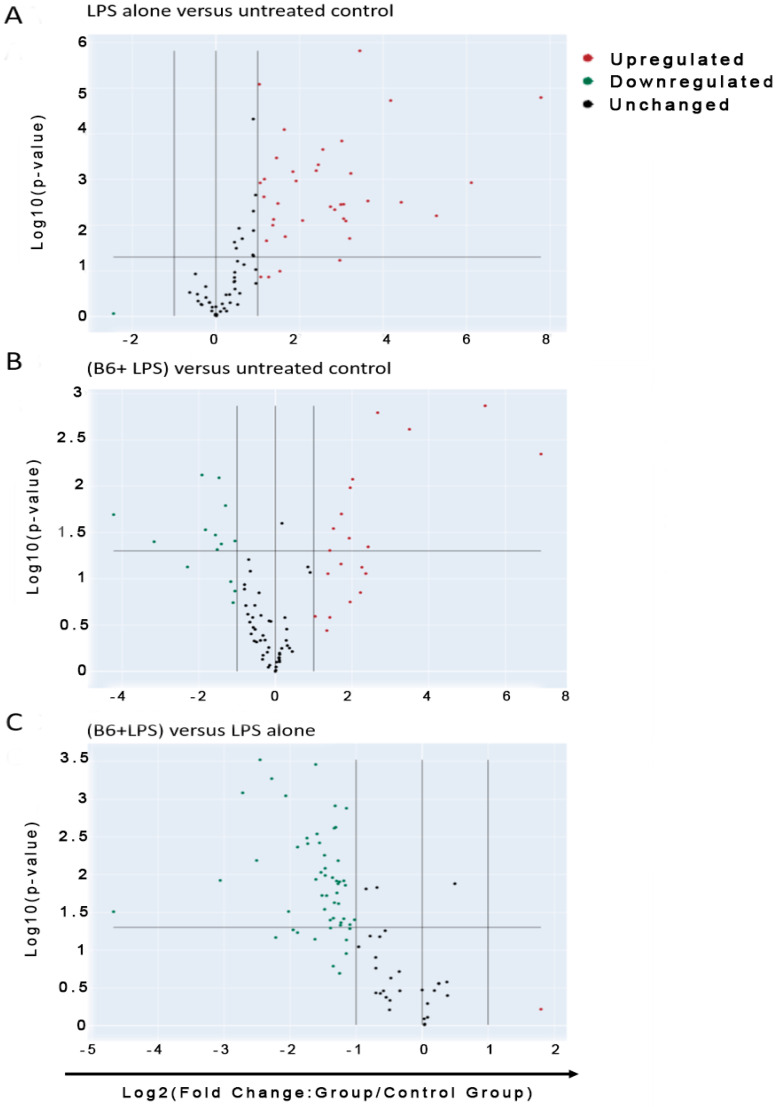
Vitamin B6 downregulates the inflammatory effect induced by LPS on monocyte/macrophage cells. Volcano plots identifying significant changes in gene expression. *x*-axis = Log_2_ fold change in gene expression. *y*-axis = statistical significance. Centre vertical line = unchanged gene expression. Outer vertical lines = selected fold regulation threshold. Horizontal line indicates *p* value threshold. Far upper left quadrant = downregulated genes of significance. Far upper right quadrant = upregulated genes of significance. U937 cells were differentiated into monocytes/macrophages and co-cultured with LPS + vitamin B6 compared to control for 24 h (*n* = 3). Cells were collected and trypsinised to ensure collection of all cells including adherent cells. mRNA was extracted and genes were profiled with RT^2^ gene profiler for innate and adaptive immune response to determine fold change in normalised gene expression compared to those that were not treated. Data analysis of (**A**) LPS treated cells vs untreated cells, (**B**) LPS +vitamin B6 treated cells compared to untreated cells and (**C**) LPS+vitamin B6 treated cells compared to LPS treated cells.

**Figure 4 biomedicines-11-02578-f004:**
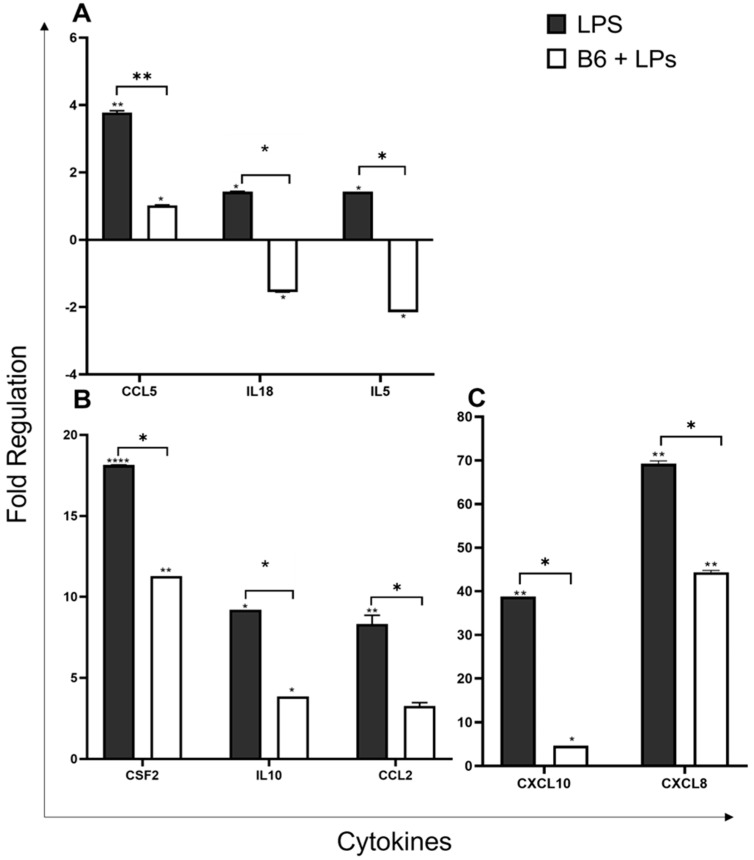
Vitamin B6 downregulates the expression of cytokines in LPS stimulated monocyte/macrophage cells. U937 cells were differentiated into monocytes/macrophages and cultured with LPS + vitamin B6 or no treatment control for 24 h (*n* = 3). Cells were trypsinised and mRNA was extracted, and genes were profiled with RT^2^ gene profiler for Innate and adaptive immune response to determine fold change in normalised gene expression compared to untreated (Control cells indicated as 0 on *x*-axis). Gene expression shown for (**A**) *CCL5, IL18, IL5,* (**B**) *CSF2, IL10, CCL2* and (**C**) *CXCL10, CXCL8*. *P* values were calculated using Student’s *t*-test where significance was determined as * *p* ≤ 0.05, ** *p* ≤ 0.01, **** *p* < 0.0001.

**Figure 5 biomedicines-11-02578-f005:**
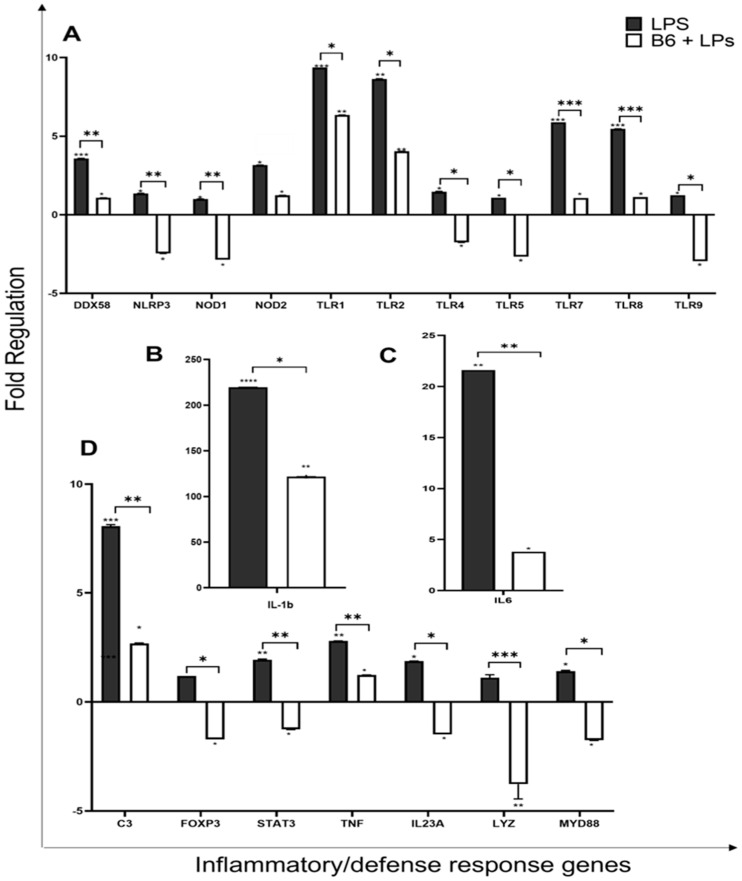
Vitamin B6 decreases the expression of inflammatory/defence response genes in LPS stimulated monocyte/macrophage cells. U937 cells were differentiated into monocytes/macrophages and co-cultured with LPS + vitamin B6 for 24 h. No treatment was used as a negative control (*n* = 3). Cells were trypsinised and collected and mRNA was extracted, and genes were profiled with RT2 gene profiler for innate and adaptive immune response to determine fold change in normalised gene expression compared to untreated; control cells were indicated as 0 on *x*-axis. Gene expression shown for (**A**) *DDX58, NLRP3, NOD1, TLR1,TLR2, TLR4, TLR5, TLR7, TLR8, TLR9,* (**B**) *IL1b*, (**C**) *IL6* and (**D**) *C3, FOXP3, STAT3, TNF, IL23A, LYZ, MYD88*. *p* values were calculated using Student’s *t*-test and significance was determined as * *p* < 0.05, ** *p* < 0.01, *** *p* < 0.001, **** *p* < 0.0001.

**Figure 6 biomedicines-11-02578-f006:**
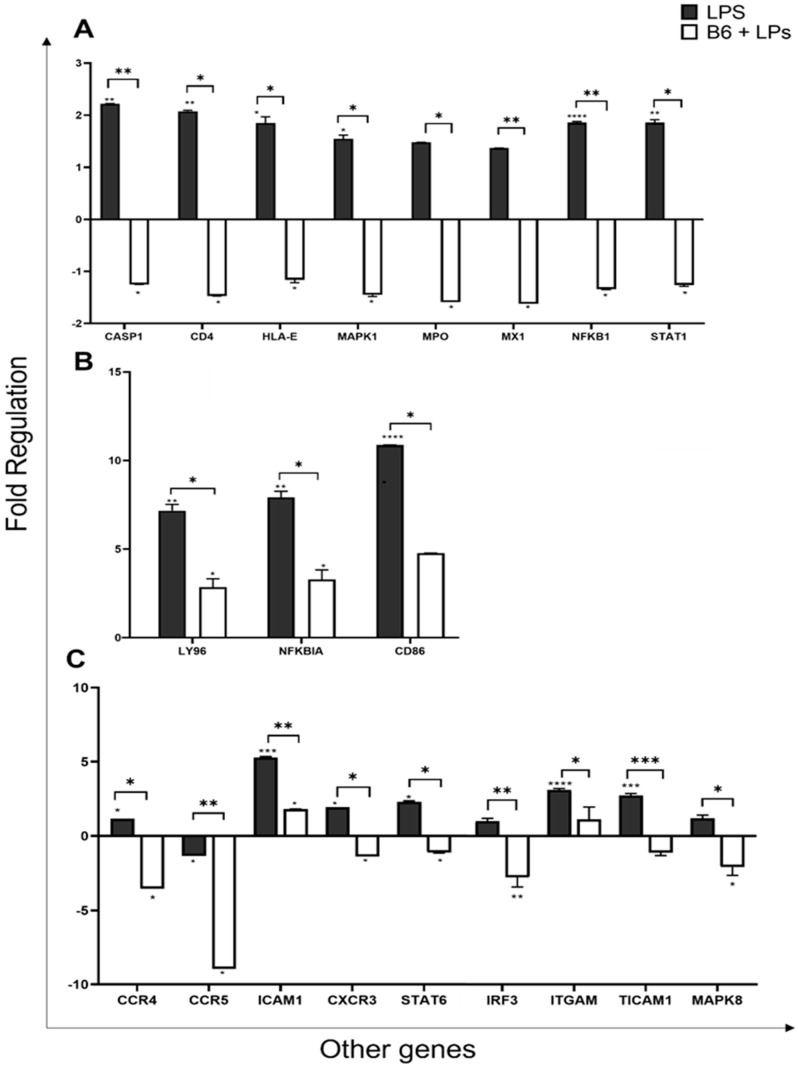
Vitamin B6 downregulates the inflammatory response of other genes in LPS stimulated monocyte/macrophage cells. U937 cells were differentiated into monocytes/macrophages and co-cultured with LPS + vitamin B6 for 24 h. No treatment was used as a negative control (*n* = 3). Cells were collected and mRNA was extracted, and genes were profiled with RT^2^ gene profiler for Innate and Adaptive immune response to determine fold change in normalised gene expression compared to untreated. Control cells are indicated as 0 on *x*-axis. Gene expression shown for (**A**) *CASP1, CD4, HLA-E, MAPK1, MPO, MX1, NFKB1, STAT1,* (**B**) *LY96, NFKB1A, CD86*, and (**C**) *CCR4, CCR5, ICAM1, CXCR3, STAT6, IRF3, ITGAM, TICAM1, MAPK8*. *p* values were calculated using Student’s *t*-test and significance determined as * *p* ≤ 0.05, ** *p* ≤ 0.01, *** *p* ≤ 0.001, **** *p* ≤ 0.0001, + *p* > 0.05.

## Data Availability

Data is available from the authors upon request.

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
