# Peer review of "High-Dose Vitamin B6 (Pyridoxine) Displays Strong Anti-Inflammatory Properties in Lipopolysaccharide-Stimulated Monocytes"

_biomedicines, 2023, doi:10.3390/biomedicines11092578_

Round 1

Reviewer 1 Report

 In this manuscript, Kathleen Mikkelsen and colleagues studied the effects of high-dose vitamin intervention on inflammation in LPS-stimulated monocytes; their finding showed a robust anti-inflammatory response, which could be helpful as a valuable therapeutic intervention in diseases associated with inflammation. This study is novel and well-organized; however, I have some minor concerns. 

In the introduction section, it would be good to give more information about the biochemistry of vitamin B6, how vitamin B6 can be utilized in the human body, molecular pathways associated with the process, etc.  

Also, please discuss how vitamin B6 is different than other anti-inflammatory compounds. 

Some references are missing-

“Further to this, vitamin B6, as a key player in one-carbon metabolism, is involved in methylation processes which, when disrupted as occurs frequently in deficiency states, can cause an increase in homocysteine resulting in vascular and systemic inflammation.”

In line 45 (introduction section), vitamin B6 can result in hyperhomocysteinemia, often associated with inflammation. Please give a reference for this statement, as this paper below studied high levels of homocysteine in mice, resulting in significant upregulation of inflammatory cytokines IL-6 and TNF-α. 

“Hydrogen sulfide alleviates hyperhomocysteinemia-mediated skeletal muscle atrophy via mitigation of oxidative and endoplasmic reticulum stress injury, AJP-Cell Physiology 2018.”

In the results section, please write how 250 ug/ml dose for vitamin 6 and 1 mg/ml dose for LPS treatment were determined for this study.

In Figure 1, please provide the calculations in bar graphs to make it more presentative. 

Please provide the primer sequences for RT-PCR analysis. 

Author Response

Reviewer 1:

In this manuscript, Kathleen Mikkelsen and colleagues studied the effects of high-dose vitamin intervention on inflammation in LPS-stimulated monocytes; their finding showed a robust anti-inflammatory response, which could be helpful as a valuable therapeutic intervention in diseases associated with inflammation. This study is novel and well-organized; however, I have some minor concerns. 

Q: In the introduction section, it would be good to give more information about the biochemistry of vitamin B6, how vitamin B6 can be utilized in the human body, molecular pathways associated with the process, etc. Also, please discuss how vitamin B6 is different than other anti-inflammatory compounds. 

A: We thank the reviewer for their comment, we have now included this information

Q: Some references are missing-

“Further to this, vitamin B6, as a key player in one-carbon metabolism, is involved in methylation processes which, when disrupted as occurs frequently in deficiency states, can cause an increase in homocysteine resulting in vascular and systemic inflammation.”

In line 45 (introduction section), vitamin B6 can result in hyperhomocysteinemia, often associated with inflammation. Please give a reference for this statement, as this paper below studied high levels of homocysteine in mice, resulting in significant upregulation of inflammatory cytokines IL-6 and TNF. “Hydrogen sulfide alleviates hyperhomocysteinemia-mediated skeletal muscle atrophy via mitigation of oxidative and endoplasmic reticulum stress injury, AJP-Cell Physiology 2018.”

A: Thank you, we have added two references

Q: In the results section, please write how 250 ug/ml dose for vitamin 6 and 1 mg/ml dose for LPS treatment were determined for this study.

A: Thank you, information added

Q: In Figure 1, please provide the calculations in bar graphs to make it more presentative. 

A: Thank you, figure 1 is now included as an additional bar graph for better presentation

Q: Please provide the primer sequences for RT-PCR analysis.

A: Thank you. The RT² Profiler PCR Array contains 84 genes related to innate and adaptive immunity, and QIAGEN includes the relevant primers in each well. We don't need to order primers separately; hence, primer sequences are not included in the method section. We checked with QIAGEN for confirmation and couldn't find any primer sequences in the kit or on their website. We have clarified this in the revised paper

Reviewer 2 Report

Excellent research work, where a precise and appropriate protocol has been carried out.

Process to make some considerations I indicate below to modify them

Monocytes in Keywords, must go in capital letters to follow the same pattern as the rest of the words.

In the abstract, LPS in paragraph 16 of article, should be specified as "lipopolysaccharides” and in brackets if desired the first time it appears (“LPS)".

In paragraph 231 “Cell surface marker expression of monocytes grown LPS ± vitamin B6”, must be bold.

In paragraph 574 (p 0.005) (Fig 6C) should end with an endpoint and a comma should go in paragraph 573 after "to -2.8-fold".

In paragraph 566 after "(p 0.005) (Fig 6C)" endpoint.

In paragraph 579 references 105 and 81 should be separated by comma and not by dot as indicated in the paper [105]. [81].

In paragraph 616 "is a potent" there appears to be a double gap.

In paragraph 674, after "has shown that the high dose of B6" there appears to be a double gap,

In paragraph 677, after "Although the potential of these findings is exciting," there seems to be a double gap. Check and if so correct it.

In paragraph 693, "in vivo" should be in italics.

The text in the bibliography is not justified.

In reference 8 missing, the volume and pages "Antioxid Redox Signal 2021, ...." doi:10.1089/ars.2021.0029. I think the volume is 36 and page 1-14.

Revise reference 18 and 30, 34, 35, 36 to complete them.

In reference 43 after "2014,," there is double comma. Delete a comma.

Generally check all references because some lack pagination.

Reference 90 is missing the DOI. doi: 10.1189/jlb.1A0415-172R, Check it!

Reference 123 missing the DOI.

Congratulations on the work done!

Author Response

Reviewer 2:

Comment: Excellent research work, where a precise and appropriate protocol has been carried out.

Process to make some considerations I indicate below to modify them

A: Thank you

Q: Monocytes in Keywords, must go in capital letters to follow the same pattern as the rest of the words.

In the abstract, LPS in paragraph 16 of article, should be specified as "lipopolysaccharides” and in brackets if desired the first time it appears (“LPS)".

In paragraph 231 “Cell surface marker expression of monocytes grown LPS ± vitamin B6”, must be bold.

In paragraph 574 (p 0.005) (Fig 6C) should end with an endpoint and a comma should go in paragraph 573 after "to -2.8-fold".

In paragraph 566 after "(p 0.005) (Fig 6C)" endpoint.

In paragraph 579 references 105 and 81 should be separated by comma and not by dot as indicated in the paper [105]. [81].

In paragraph 616 "is a potent" there appears to be a double gap.

In paragraph 674, after "has shown that the high dose of B6" there appears to be a double gap,

In paragraph 677, after "Although the potential of these findings is exciting," there seems to be a double gap. Check and if so correct it.

In paragraph 693, "in vivo" should be in italics.

The text in the bibliography is not justified.

In reference 8 missing, the volume and pages "Antioxid Redox Signal 2021, ...." doi:10.1089/ars.2021.0029. I think the volume is 36 and page 1-14.

Revise reference 18 and 30, 34, 35, 36 to complete them.

In reference 43 after "2014," there is double comma. Delete a comma.

Generally, check all references because some lack pagination.

Reference 90 is missing the DOI. doi: 10.1189/jlb.1A0415-172R, Check it!

Reference 123 missing the DOI.

A: All minor edits added in the revised paper

Comment: Congratulations on the work done!

A: We thank the reviewer